



**Physical Processes Leading to Extreme Day-to-day Temperature Change - Part II: Future Climate Change**

Kalpana Hamal and Stephan Pfahl

Institut für Meteorologie, Freie Universität Berlin, 12165 Berlin., Germany

Correspondence to: Kalpana Hamal (*k.hamal@fu-berlin.de*)

**Abstract**

Extreme temperature swings from one day to the next, whether warming or cooling, can significantly impact human health, ecosystems, and the economy. These effects may become more pronounced or attenuated in the future. Part I of this research identified the physical processes—advection, as well as adiabatic and diabatic temperature changes—that cause extreme day-

to-day temperature (DTDT) fluctuations in the present climate. However, how these processes influence the projected extreme DTDT change under warming scenarios remains unknown. This study addresses this question globally by analysing physical processes in Community Earth System Model Large Ensemble (CESM-LE) simulations under a high-emission scenario, employing both Eulerian composite and Lagrangian backwards-trajectory analyses. The projected changes in (extreme) DTDT variations display a clear seasonal contrast with a dipole pattern: weakening in mid- to high latitudes and intensification in the

tropics during December–February (DJF), while during June–August (JJA), tropical intensification is more widespread, and only some extratropical locations experience reductions in DTDT variations. The projected changes in the magnitude of DTDT variations are mostly linked to changes in the standard deviation of daily temperature, while changes in the temporal autocorrelation also play a role in some regions. In the extratropics during DJF, the weakening of DTDT extremes is mainly driven by reduced advection contributions due to Arctic amplification. However, during JJA, reductions in extremes result

from changes in advection, diabatic, and adiabatic processes, with differences between events and regions in their relative contributions. Furthermore, changes in diabatic processes play a significant role in the projected intensification of extremes in JJA over land areas in the tropics and subtropics, while the tropical intensification during DJF results from local changes in diabatic and adiabatic processes. Our findings demonstrate that a regional and seasonal perspective that, in addition to the well-established role of changes in advection, also accounts for diabatic and adiabatic heating processes, is essential for

understanding projected extreme DTDT changes and for developing suitable adaptation strategies.

## 1. Introduction

In its Sixth Assessment Report, the Intergovernmental Panel on Climate Change emphasises that temperature extremes have adverse effects on human health, agriculture, and the economy, and anticipates these challenges worsening as associated weather extremes intensify (Intergovernmental Panel on Climate, 2023). Rapid day-to-day temperature (DTDT) changes, as

one specific type of temperature extremes (Hamal & Pfahl, 2025), significantly impact health, primarily contributing to increased mortality rates, especially among children and older individuals (Chan et al., 2012; Hovdahl, 2022; Martínez-Solanas



et al., 2021; Wu et al., 2022). They also lead to economic losses, which are notably higher at low latitudes than at high latitudes, and negatively affect agriculture (Kotz et al., 2021; Linsenmeier, 2023; Zou et al., 2024). Furthermore, these impacts are projected to increase in a warming future, particularly affecting economic activity in warm, poor regions (Linsenmeier, 2023) and being associated with reductions in cropland and yields (Wang et al., 2022). The population's exposure to the risk of these DTDT changes has continued to increase (Chen et al., 2025). Therefore, studying DTDT changes and their extremes in a warming climate is imperative.

Research on future projected changes in DTDT variations and their extremes remains limited. Zhou et al. (2020) projected a decrease in the magnitude of extreme DTDT variations across mid to high latitudes on the annual time scale, associated with declining strong wind patterns. Similar results have been observed for the Northern Hemisphere winter and summer (Kim et al., 2013; Wang et al., 2025). Xu et al. (2020) found a reduction in DTDT variations during winter and an increase along the Arctic Coast during summer, driven by notable shifts in the meridional temperature gradient. Similarly, an increase in summer variability across extratropics and tropical landmasses has been attributed to anthropogenic influences (Wan et al., 2021). All these studies have examined the typical or average magnitude and trends of DTDT changes, except for Zhou et al. (2020) and Liu et al. (2025), who also examined extreme DTDT changes using a fixed temperature threshold (Zhou et al., 2020) and a percentile threshold method (Liu et al., 2025), respectively, on an annual timescale. Here, to investigate projected future extreme DTDT changes at a seasonal timescale, a percentile-based threshold method is applied, similar to previous studies of temperature extremes in a warming climate (Schielicke & Pfahl, 2022; Vogel et al., 2020). Furthermore, we aim to investigate the detailed thermodynamic and dynamic processes underlying the projected changes in extreme DTDT variations using Eulerian composites and Lagrangian backward trajectories.

Backward trajectory analyses have been widely used to identify the physical processes underlying the formation of temperature extremes, encompassing advection (the transport of air from climatologically warmer regions to colder regions or vice versa), Lagrangian temperature changes associated with adiabatic compression or expansion, and diabatic heating or cooling. Such process-based analyses have been conducted both in the past (Bieli et al., 2015; Mayer, 2025; Nygård et al., 2023; Papritz & Spengler, 2017; Quinting & Reeder, 2017; Röthlisberger & Papritz, 2023a, 2023b; Zschenderlein et al., 2019) and within the context of future climate change (Brunner et al., 2018; Chan et al., 2022; Jeong et al., 2025; Schaller et al., 2018; Schielicke & Pfahl, 2022), providing a better understanding of temperature extremes. These studies also highlighted that temperature extremes are intricately linked to synoptic-scale circulation patterns, such as ridges and troughs, which control the advection of air masses and the adiabatic warming or cooling due to subsidence or ascent, respectively (Jeong et al., 2025; Kautz et al., 2022; Neal et al., 2022). Additionally, turbulent mixing and diabatic processes, such as radiative cooling and sensible heat fluxes near the Earth's surface, significantly contribute to the formation of extreme temperatures (Hartig et al., 2023; Mayer, 2025; Röthlisberger & Papritz, 2023a, 2023b). Future changes in extreme temperatures (daily temperature extremes and heatwaves) are primarily driven by thermodynamic processes, with less influence from dynamic processes (Brunner et al.,



2018; Chan et al., 2022; Schaller et al., 2018; Schielicke & Pfahl, 2022; Vogel et al., 2020). Here, we investigate whether this also holds for extreme DTDT changes in a warming climate.

Building on the methodology and processes understanding from Part I of this study (Hamal & Pfahl, 2025), we investigate
historical and future extreme DTDT changes in global climate simulations using Lagrangian backward trajectory analyses of surface air masses initialised at selected locations on the two days involved in extreme DTDT changes. Furthermore, the contributions of various processes—advection, adiabatic, and diabatic warming/cooling—to projected future extreme DTDT changes are analysed using a Lagrangian temperature decomposition. This study aims to address the following research questions: (1) What is the role of changes in atmospheric circulation patterns for projected future changes in extreme DTDT
variations? (2) Which physical processes contribute to extreme DTDT changes in a warming climate?

## 2. Data and methodology

### 2.1 CESM-LE

In this study, we use 30 ensemble members from the fully coupled Community Earth System Model Large Ensemble (CESM-LE) project (Kay et al., 2015) to cover the influence of natural variability. The ensemble members differ by small random
perturbations applied to their initial air temperature fields, with magnitudes of approximately $10^{-14}$ K. The simulations are externally forced using historical conditions up to 2005 and representative concentration pathway (RCP) 8.5 conditions for 2006–2100. The atmospheric variables in the CESM-LE dataset are available on a horizontal grid with approximately 1 degree spacing in latitude and 1.25 degrees in longitude, with 30 hybrid vertical levels and 6-hourly intervals. Our analysis focuses on two 10-year time slices: 1991–2000 (historical climate) and 2091–2100 (future climate). The CESM simulations were rerun
for these time slices using restart files from the original CESM-LE simulations to generate additional output required for the trajectory calculations (Dolores-Tesillos et al., 2022; Schielicke & Pfahl, 2022). Afterwards, all fields were remapped to a uniform horizontal resolution of 1° × 1°. The analysis incorporates near-surface temperatures at a reference height of 2 meters above ground level, total cloud cover, precipitation, and several three-dimensional atmospheric fields, including temperature, pressure, geopotential height, and horizontal and vertical wind components. The temporal resolution of near-surface
temperature and composite analyses is daily (averages calculated from 6-hour intervals), whereas the input data for trajectory calculations are maintained at 6-hour resolution. Previous applications of the CESM-LE for simulating temperature extremes and associated processes across various regions demonstrate its reliability and confirm its suitability for the present study (Schielicke & Pfahl, 2022; Wang et al., 2019).

### 2.2 ERA5

We utilise 2m air temperature (calculated from hourly data) data from 1980 to 2020, at a spatial resolution of 1° × 1° from the fifth-generation European Centre for Medium-Range Weather Forecasts (ECMWF) global reanalysis product (ERA5,





(Hersbach et al., 2020)), to calculate present-day DTDT variability and compare it with the model simulations. More details are provided in Section 2 of Part I (Hamal & Pfahl, 2025).

## 2.3 Calculation of DTDT changes and their extremes

The DTDT change, $\delta_T$, is defined as the difference in daily mean near-surface air temperature between the day of the event ($t$) and the previous day ($t$-1). Here, $T_{t-1}$ and $T_t$ represent the near-surface air temperatures on these two days, respectively. The standard deviation of the DTDT change $\sigma_{DTDT}$ can be expressed as a function of the usual standard deviation $\sigma_T$ and the lag-1 autocorrelation $r_{1,T}$ of the daily mean temperature, as indicated in Eq. (1), which was derived in Section 2 of Part I (Hamal & Pfahl, 2025). We calculate all these quantities for the historical and future scenarios for each ensemble member, then determine

the differences between the scenarios for each member and the ensemble mean.

$$\sigma_{DTDT} = \sigma_T\sqrt{2\ (1\text{-}r_{1,T})} \qquad (1)$$

Extreme DTDT changes are examined using the percentile method for both historical and future climates. Cooling and warming events are determined at each grid point and for each ensemble member, using the 5th and 95th percentiles of DTDT

change as thresholds. The analysis focuses on two key seasons: December–February (DJF) and June–August (JJA). At each location, 44 events are selected for DJF and 45 for JJA for each member and each ten-year time slice (historical and future climate).

## 2.4 Trajectory calculation

We employ a Lagrangian analysis method to compute backwards trajectories, similar to the approach used for ERA5 in (Hamal

& Pfahl, 2025), but here applied to each CESM-LE's historical and future extreme DTDT changes during both days $t$-1 and $t$, respectively. 3-day backward trajectories are initialised at 18 UTC from pressure levels of 10, 30, 50, and 100 hPa above the surface at each selected grid box (see section 2.5). Output data, including latitude, longitude, pressure, temperature, and potential temperature, are recorded at 6-hour intervals. We then compare the trajectories for extreme events in the historical and future scenarios to assess their projected changes.

## 2.5 Lagrangian temperature decomposition

To better understand the underlying mechanisms of extreme DTDT changes, our analysis focuses on different locations that show significant projected future changes during DJF and JJA. For DJF, we select two locations: North America (51°N, 86°W) and tropical South America (13°S, 56°W), whereas for JJA, we focus on western North America (45°N, 120°W) and central Europe (50°N, 10°E) in the main paper. Additional grid points in Northern Asia (70°N, 90°E), Northern Europe (60°N, 10°E),

tropical Southern Africa (13°S, 24°E) for DJF, eastern North America (41°N, 76°W), Southern Asia (35°N, 80°E), the Sahel region (15°N, 5°E), Southern South America (47°S, 70°W), Northern Asia (68°N, 80°E) and Southern South Africa (30°S, 22°W) for JJA are shown in the supplementary material. At these locations, the Lagrangian decomposition method (eq. 2)



developed in Part I (Hamal & Pfahl, 2025) is utilised to quantify the contributions of advection, adiabatic, and diabatic processes to extreme DTDT changes in both historical and future scenarios. This method is applied to 3-day backward trajectories initiated during the two days associated with extreme DTDT change events in both present and future climates.

$$\delta_T^0 \approx \delta_{\overline{T}}^{-3d} + \delta_{\overline{T}}^{adi} + \delta_{\overline{T}}^{dia} + res \qquad (2)$$

Here, the DTDT change ($\delta_T^0$) is decomposed into three contributing factors: the mean temperature difference at the origin of the air masses three days before initialization, which indicates the contribution of changes in advection change ($\delta_{\overline{T}}^{-3d}$), a contribution of mean adiabatic compression or expansion change resulting from vertical descent or ascent ($\delta_{\overline{T}}^{adi}$), and a contribution of mean diabatic heating or cooling change from processes such as latent heating in clouds, radiation, and surface fluxes ($\delta_{\overline{T}}^{dia}$). The final term is the residuum change (*res*), resulting from numerical inaccuracies in the calculations. The residual change is usually small and therefore not further addressed in the subsequent text and figures. Equation (2) is applied separately for the historical and future climate, and projected changes in the contributions are calculated as differences between the time slices.

## 3. Result

### 3.1 Projected future DTDT changes in DJF and JJA

During the historical climate period, temperature variability in the CESM-LE simulations, quantified through both $\sigma_{DTDT}$ and $\sigma_T$, is typically larger in the mid-to-high latitudes of both the Northern and Southern Hemispheres compared to the tropics. This pattern is consistently observed during DJF and JJA (Figures 1a-b and 2a-b). The model results lie within ±10% of estimates based on ERA5 (cf. Part I) in many regions (stippling in Figures 1a-b and 2a-b), which enhances confidence in its projections. Notable exceptions include large parts of the Northern Hemisphere mid- and high latitudes and Southeast Asia during DJF (Figures 1a-b), where the model overestimates variability, and several subtropical regions during JJA (Figures 2a-b and Fig. S1). The simulated autocorrelation generally agrees well with ERA5 (Figures 1c, 2c), though there are regional exceptions, notably in the deep tropics, Central America, and the Arabian Peninsula. Furthermore, also in CESM-LE, the spatial pattern of $\sigma_{DTDT}$ is generally determined by $\sigma_T$ (as outlined in Eq. 1), with influences of $r_{1,T}$ being restricted to the regional scale, as discussed for ERA5 in Part I.





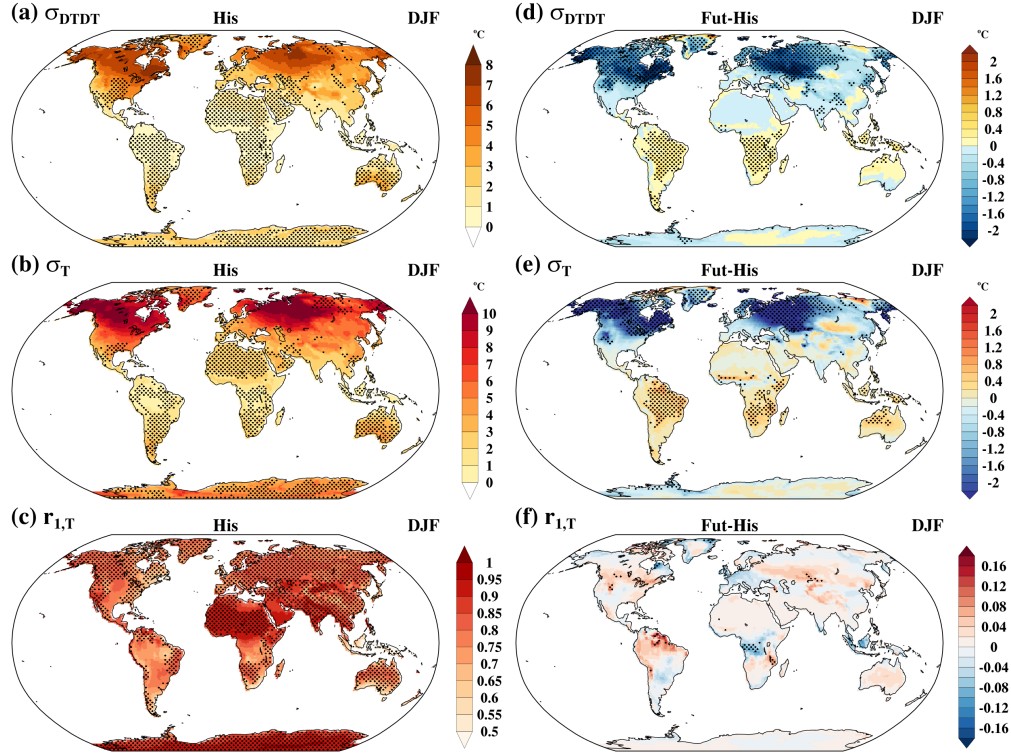

**Figure 1.** The ensemble means of **(a, d)** standard deviation of DTDT variations ($\sigma_{DTDT}$, °C), **(b, e)** standard deviation of daily mean temperature ($\sigma_T$, °C), and **(c, f)** lag-1 autocorrelation of daily mean temperature ($r_{1,T}$) in December-February (DJF) in the historical climate **(a-c)** and projected future change **(d-f)**. Stippling indicates locations where at least 80% of CESM-LE members lie within ±10% of the ERA5-derived respective quantities, indicating model consistency with observations. Meanwhile, for the projected future change, stippling indicates areas where ≥80% of ensemble members agree on the sign of change.

The projected future changes in $\sigma_{DTDT}$, $\sigma_T$, and $r_{1,T}$ compared to the historical climate are illustrated in Figures 1d-f and 2d-f. During DJF, projected changes of $\sigma_{DTDT}$ and $\sigma_T$ exhibit a distinct dipole pattern (Figures 1d-e). This pattern indicates a significant decrease in the magnitude of DTDT variations over the northern mid-to-high latitudes, where the spread between ensemble members is also largest (locally above 0.6°C, Fig. S2). In contrast, an increase in variability is projected in tropical regions such as the Amazon, southern Africa, the Maritime Continent, and northern Australia. The pronounced changes $\sigma_{DTDT}$ are primarily driven by changes in $\sigma_T$, as $r_{1,T}$ does not exhibit significant changes (Figure 1f), with a few regional exceptions such as an area in tropical western Africa, where $\sigma_{DTDT}$ increases due to a reduction of $r_{1,T}$ in spite of minor changes in $\sigma_T$ (see again Eq. 1).



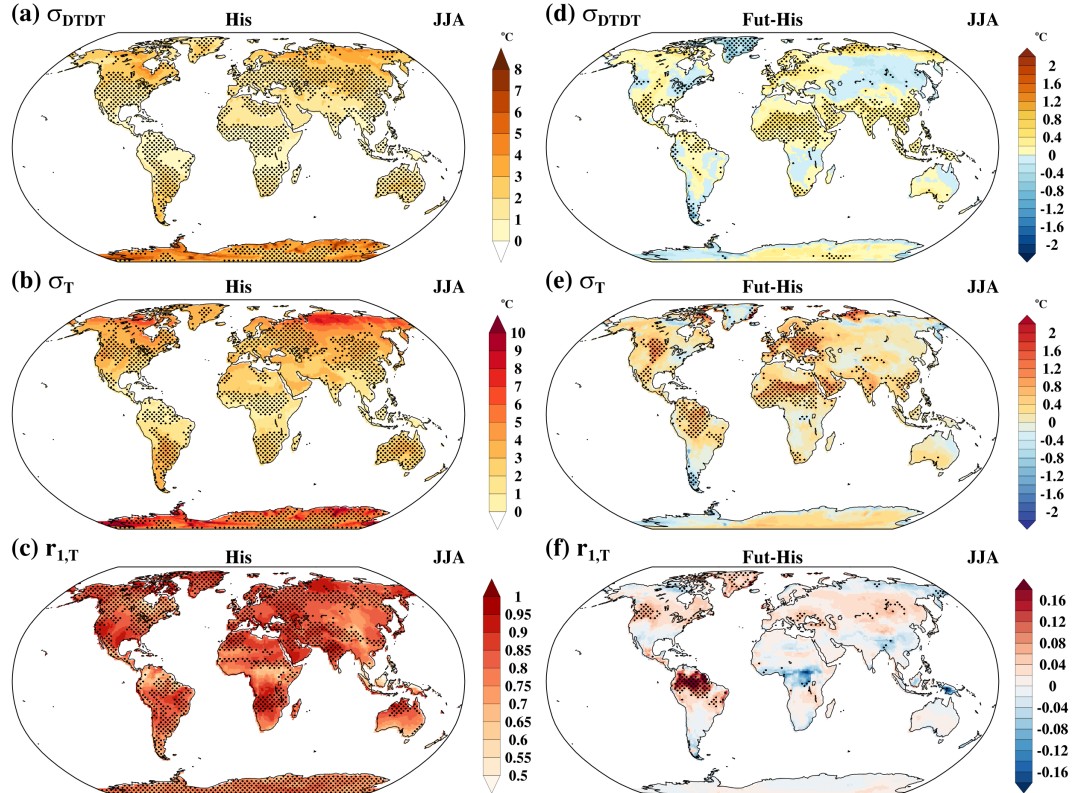

**Figure 2.** The ensemble means of **(a, d)** standard deviation of DTDT variations ($\sigma_{DTDT}$, °C), **(b, e)** standard deviation of daily mean temperature ($\sigma_T$, °C), and **(c, f)** lag-1 autocorrelation of daily mean temperature ($r_{1,T}$) in June-August (JJA) in the historical climate **(a-c)** and projected future change **(d-f)**. Stippling indicates locations where at least 80% of CESM-LE members lie within ±10% of the ERA5-derived respective quantities, indicating model consistency with observations. Meanwhile, for the projected future change, stippling indicates areas where ≥80% of ensemble members agree on the sign of change.

During JJA, most regions globally are projected to experience either minor changes or an increase in both $\sigma_{DTDT}$ and $\sigma_T$, and $\sigma_{DTDT}$ is thus strongly influenced by $\sigma_T$. This pattern is primarily observed in regions such as Europe, the Amazon, the Sahel, the Maritime Continent, South and Southeast Asia, Central America, and parts of northern Russia (Figures 2d–e). However, exceptions include western and eastern North America, Greenland, and Chile, where $\sigma_{DTDT}$ is expected to decrease despite no significant changes in $\sigma_T$ (apart from Chile). This discrepancy can be attributed to a substantial future increase in $r_{1,T}$, particularly over western North America and Greenland (Figure 2f). The magnitude of the response is rather similar across the ensemble for JJA, in contrast to the larger variations observed for DJF in the Northern Hemisphere (Fig. S2).

Figure S3 illustrates the statistical distributions of historical and projected future DTDT changes for selected locations, including North America, central Europe, tropical South America, and western North America (cf. Section 2.5). During DJF and JJA, North American regions experience the highest variability in DTDT changes, characterised by broad distributions in




the historical climate (Fig. S3a, c). However, this variability is expected to decrease in the future, with the distribution becoming narrower and exhibiting a higher peak. In contrast, tropical South America has the lowest historical variability, with a sharp, pronounced peak that broadens (increased SD and reduced kurtosis) in the future during DJF (Fig. S3b). In central

Europe during JJA, the changes are relatively minor, yet there is an increase in standard deviation and kurtosis (Fig. S3d). Tropical South America and central Europe are thus expected to become more variable in terms of DTDT changes, while the variability is projected to decrease over North American regions, in accordance with the previously discussed changes $\sigma_{DTDT}$.

## 3.2 Projected extreme DTDT changes

The 5th and 95th percentiles serve as thresholds at each grid point to identify historical and projected future extremes of DTDT

cooling and warming, as depicted in Figure 3. The future patterns of extreme DTDT changes closely correspond with the projected climatological $\sigma_{DTDT}$ and $\sigma_T$ patterns (Figures 1, 2, and 3). During DJF, the areas where historical extreme DTDT changes have the highest magnitude, primarily in mid- to high latitudes, are expected to experience a decrease in intensity in the future (Figures 3e-f). In contrast, in tropical regions, these extremes are projected to intensify. In JJA, projected future changes exhibit regional variations, with intensities mainly increasing in parts of the tropics and subtropics. Exceptions with

weakening extreme DTDT changes are observed in some extratropical regions, including western and eastern North America, Greenland, and Chile (Figures 3g-h).

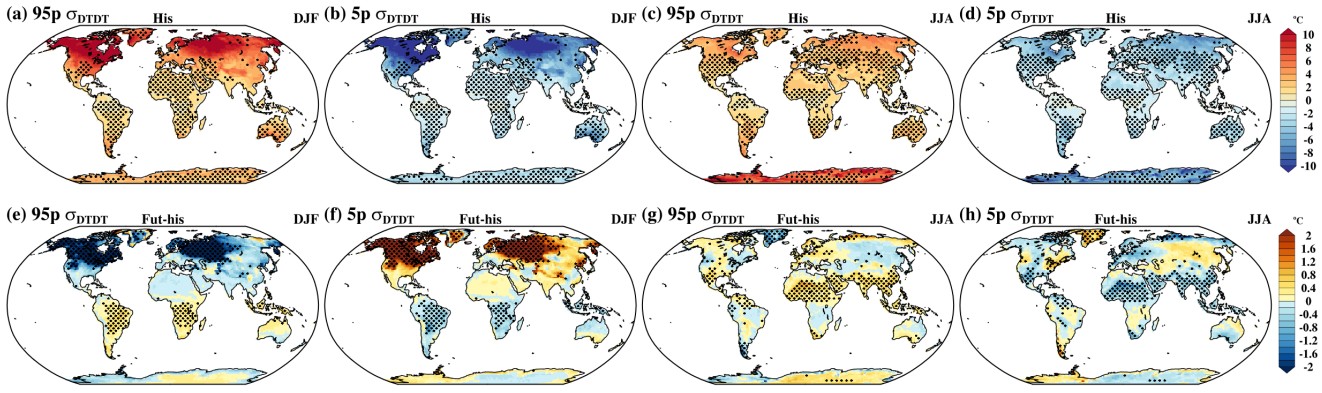

**Figure 3.** The ensemble means of the **(a, e, c and g)** 95th percentile (95p) and **(b, f, d and h)** 5th percentile (5p) of DTDT variations during **(a-b, e-f)** December-February (DJF) and **(c-d, g-h)** July-August (JJA) based on the historical climate **(a-d)** and projected future change **(e-**
**h).** Stippling indicates locations where at least 80% of CESM-LE members lie within ±10% of the ERA5-derived respective quantities, indicating model consistency with observations. Meanwhile, for the projected future change, stippling indicates areas where ≥80% of ensemble members agree on the sign of change.

## 3.2.1 Projected future weakening of extreme DTDT changes

During DJF, the northern mid- to high-latitude regions (Northern America, Northern Asia, and Northern Europe) are projected

to experience a decrease in extreme DTDT changes compared to the historical climate, driven by consistent changes in




atmospheric circulation and physical processes, as illustrated in Figures 4-6 and Figs. S4-6. This section focuses exclusively on the results for North America. Composites of anomalies from the seasonal climatology are shown for both days involved in an extreme DTDT change for geopotential height (GP) at 500 hPa, wind at 850 hPa, and sea level pressure (SLP) in Figure 4, together with changes in the day-to-day circulation differences in Fig. S4.


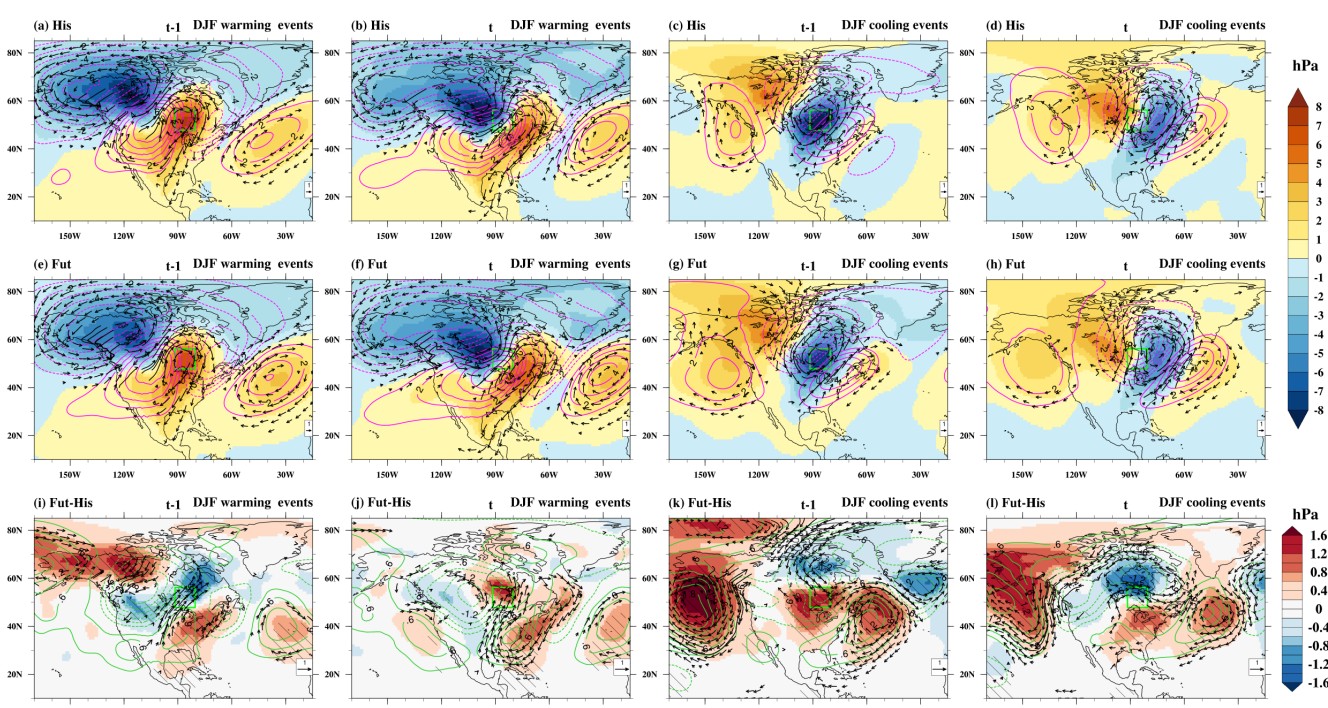

**Figure 4**. Composite of sea level pressure anomalies (hPa, color shading), wind anomalies at 850 hPa (**m s$^{-1}$**, vectors), and geopotential height anomalies at 500 hPa (gpm, magenta and darkgreen contours) relative to the seasonal mean on the **(a, e, i, c, g, k)** previous day (**t-1**) and **(b, f, j, d, h, l)** the event day (**t**) of the warming **(a-b, e-f and i-j)** and cooling **(c-d, g-h and k-l)** events during December-February (DJF) in **(a-d)** historical climate (His), **(e-h)** future climate (Fut), and **(i-l)** projected future changes (Fut-His) at a selected grid box in North America (green box). Note that, in **(a-h)**, wind vector anomalies ≥ 2 **m s$^{-1}$** and in **(i-l)**, wind vector difference anomalies ≥ 0.5 **m s$^{-1}$** are plotted. The dotted and bold contours indicate negative and positive geopotential height anomalies, respectively. Additionally, the hatched area indicates where the ensemble mean of sea level pressure differences exceeds the 95% confidence threshold based on a Student's t-test.

In the simulated historical period, DTDT warming events are associated with pronounced shifts in GP, SLP, wind anomalies, and air mass origin between days *t-1* and *t* (Figures 4a–b and 5a–b). On day *t-1*, the grid box lies within a high-pressure area downstream of an eastward-propagating ridge at 500 hPa, in a transition zone between southwesterly wind anomalies to the west and northeasterlies to the east (Figure 4a), associated with the advection of relatively cool air masses (Figures 5a, m). By day *t*, the ridge moves over the grid box, featuring southwesterly wind anomalies in between an upstream surface low and the high that shifted downstream (Figure 4b) and the advection of much warmer continental air masses (Figures 5b, m). This shift to southwesterly advection (see also Fig. S4a) contributes an average warming of +6.2°C to the DTDT change (Figures 5m





and 6a). In terms of vertical transport, weaker subsidence at *t* compared to *t-1* reduces the adiabatic warming by –2.6°C (Figures 5i, 6a), partially offsetting the temperature increase. Conversely, enhanced diabatic heating during the final 24 hours of the event day contributes +4.8°C to the DTDT change (Figure 5q), further amplifying the warming. The strength of these process

contributions varies across events, as illustrated by the box-and-whisker plots (Figure 6a). In summary, the simulated DTDT warming events during the historical period are primarily driven by a shift toward southwesterly warm-air advection, with diabatic heating reinforcing the effect and adiabatic changes slightly dampening it (Figure 6a). These model results generally align with the patterns and process contributions identified from ERA5 reanalysis data (see Figures 4a-c and 5k in Part I). A quantitative comparison reveals that CESM-LE indicates a slightly smaller advection contribution but a larger diabatic

contribution than ERA5. Nonetheless, the overall agreement between the model and reanalysis results enhances confidence in the future projections derived from CESM-LE, as discussed below.

In general, the synoptic-scale flow pattern associated with future DTDT warming events (Figure 4e-f) is similar to the historical pattern. However, there are signals that SLP and GP anomalies are predicted to weaken and shift southeastward relative to the

historical period (Figure 4i–j), indicating a more rapid Rossby wave propagation. This faster propagation leads to an earlier shift to southwesterly flow toward the selected location, resulting in southwesterly wind anomalies and a southwestward change of air mass origins relative to the historical simulation at *t-1* (Figures 4i, 5e). On day *t*, the further downstream location and slight change in the orientation of the GP anomalies (Figure 4j) are associated with an enhanced meridional flow of southerly air masses into the region (Figures 4j, 5f). Hence, on both days involved in future DTDT warming events, the flow

has a stronger southerly component. Still, the strengthening of the southwesterlies is more pronounced at *t-1*; the difference between the two days thus shows a northeasterly anomaly (Fig. S4c). These relatively subtle changes in circulation patterns are associated with an average net reduction of –1.1°C in the advective contribution to DTDT warming compared to the historical period (Figure 6c). This reduction is because the future warming at the origin of the traced air masses (at -3d) is 9.4°C at *t-1* but only 8.3°C at *t* (Figure 5n). In addition to changes in the wind (with stronger southerlies in particular at *t-1*),

future changes in the meridional temperature gradient may contribute to this differential warming, with Arctic amplification being expected to lead to larger warming in the still more northerly source regions at *t-1* compared to *t* (see Figures 4i-j and 5e-f). Projected changes in vertical motion indicate slightly increased subsidence on day *t-1* (by 5–6 hPa in 3d, Figure 5j) compared to day *t*, which modestly enhances adiabatic warming, reducing the warming between *t-1* and *t* by an additional –0.5°C. Diabatic heating decreases relative to the historical period on both days, with a slightly larger reduction on the event

day (mainly in the last 24h before arrival; Figure 5r), contributing an additional –0.5°C to the overall decrease in warming. Overall, the weakening of all three contributions together results in a mean reduction in the DTDT warming of –2.1°C, with the reduced advective contribution accounting for approximately half of this signal. However, changes in diabatic and adiabatic warming also play a significant role (Figure 6c).







**Figure 5.** The spatial distribution of trajectory density initiated on the previous day (***t-1***) and the event day (***t***) is depicted for both December-February (DJF) warming and cooling events over North America (green box). Color shading shows the air mass density (%) during the 3d before arriving at the target grid box for **(a-d)** historical climate and **(e-h)** projected future change. In **(e-h)**, stippling indicates areas where ≥80% of the ensemble members agree on the sign of change. The mean Lagrangian evolution of distinct physical parameters (pressure, temperature, and potential temperature) is shown along the air mass trajectories initialised on the previous and event days for historical/future extreme events (1st and 3rd columns) and projected future changes in extremes (2nd and 4th columns). Additionally, bold circles show where the ensemble mean differences at each timestep exceed the 95% confidence threshold based on a Student's t-test.





During historical DTDT cooling events, a southeastward transition of a trough-ridge pattern and associated SLP anomalies
between *t-1* and *t* dominates the temperature drop (Figures 4c-d, 5c-d, and Fig. S4b). Specifically, the shift from a
southwesterly advection of a warm air mass downstream of a low-pressure anomaly on day *t-1* to northerly transport of a
colder air mass between an upstream surface high and the downstream low on day *t* drives the cooling. This pattern results in
a large advective contribution of –9.8°C to the temperature decrease (Figures 5o and 6b). The advective cooling is partially
offset by moderate adiabatic warming (+2.9°C), resulting from the enhanced subsidence and descent of air masses on the event
day (Figures 5k and 5o). Additionally, diabatic cooling on day *t* contributes –2.4°C (Figure 5s), further intensifying the overall
temperature decline. Collectively, the strong cold-air advection, reinforced by diabatic cooling, is the primary driver of DTDT
cooling events across North America and other mid- to high-latitude regions (Figures 6b and Figs. S5–S6). A comparison of
North American cooling events reveals that, while the diabatic cooling is similar between CESM-LE and ERA5, the cold air
advection is slightly stronger in CESM-LE (see Figures 4d-f and 5l, in Part I).


The synoptic circulation characterising future DTDT cooling events is very similar to the historical events (Figures 4c-d, g-h,
k-l, and Fig. S4d). On both days, a slight northward shift of the low-pressure anomaly is associated with (north-) westerly wind
anomalies in the selected grid box (Figures 4k, l) and a westward shift of the trajectory origins (Figures 5g, h). Nevertheless,
the average temperature of the air masses three days before their arrival increases by 10.3°C on the day of the event, compared
to a warming by only 7.5°C for the trajectories initialized at *t*-1, which is likely due to the generally larger warming at higher
latitudes, where the air masses on the day of the event originate, associated with Arctic amplification (Figure 5p). As a result,
the contribution of advective cooling is substantially reduced in future events relative to the historical period (+2.8°C, Figure
6d). Vertical motion changes also show slightly enhanced subsidence at *t* (approximately 4–5 hPa in 3d; Figure 5l), which
contributes to a modest increase in adiabatic warming (+0.4°C) and further limits surface cooling. Diabatic cooling is similarly
weaker in the future climate, with a slightly greater reduction on day *t-1*—particularly during the final 24 hours (Figure 5t)—
leading to an additional change of +0.4°C. Overall, the dominant factor limiting future DTDT cooling (+3.6°C) is the reduced
strength of cold air advection, likely associated with Arctic amplification, with smaller but reinforcing contributions from
enhanced adiabatic warming and reduced diabatic cooling (Figure 6d).



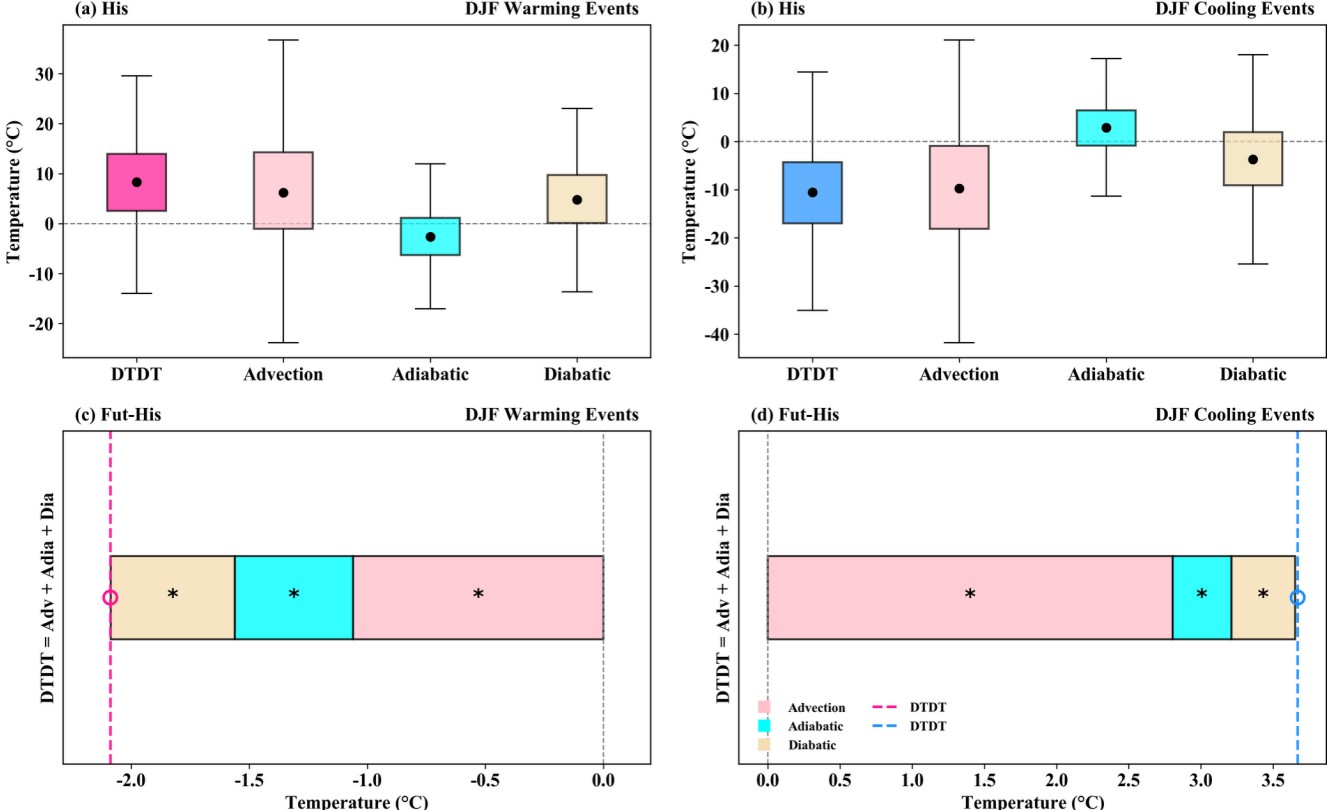


**Figure 6.** The contribution of the different physical processes (advection, adiabatic and diabatic temperature change) over North America during December-February (DJF) to genesis of DTDT **(a, c)** warming and **(b, d)** cooling events during historical climate **(a-b, box plots)** and projected future change **(c-d, stacked plots)** according to Eq. (2), which refers to a 3d-time scale. The box spans the 25th and 75th percentiles of the trajectory data; the black dot inside the box gives the mean of the related quantities in the historical climate, and the whiskers indicate 1.5 times the interquartile range in panels **(a)** and **(b).** The dotted lines in the stacked plots in panels **(c)** and **(d)** show the mean future change for DTDT warming and cooling events**,** respectively, and coloured bars indicate the contributions of the individual processes. Circle and * symbols mark future change distributions for which the ensemble mean differences exceed the 95% confidence threshold based on a Student's t-test.


The CESM-LE future projections suggest that extreme DTDT changes will also weaken during JJA in some extratropical regions, including eastern and western North America, Greenland, and Chile. Regarding the underlying processes, the results differ from those of extreme DJF events, with only a weakening advective contributing to this reduction (Fig. S8). Additionally, changes in diabatic (Fig. S9) or adiabatic processes may also occur. As an example, a grid box in western North America is examined in detail in the following section (Figures 7-9).






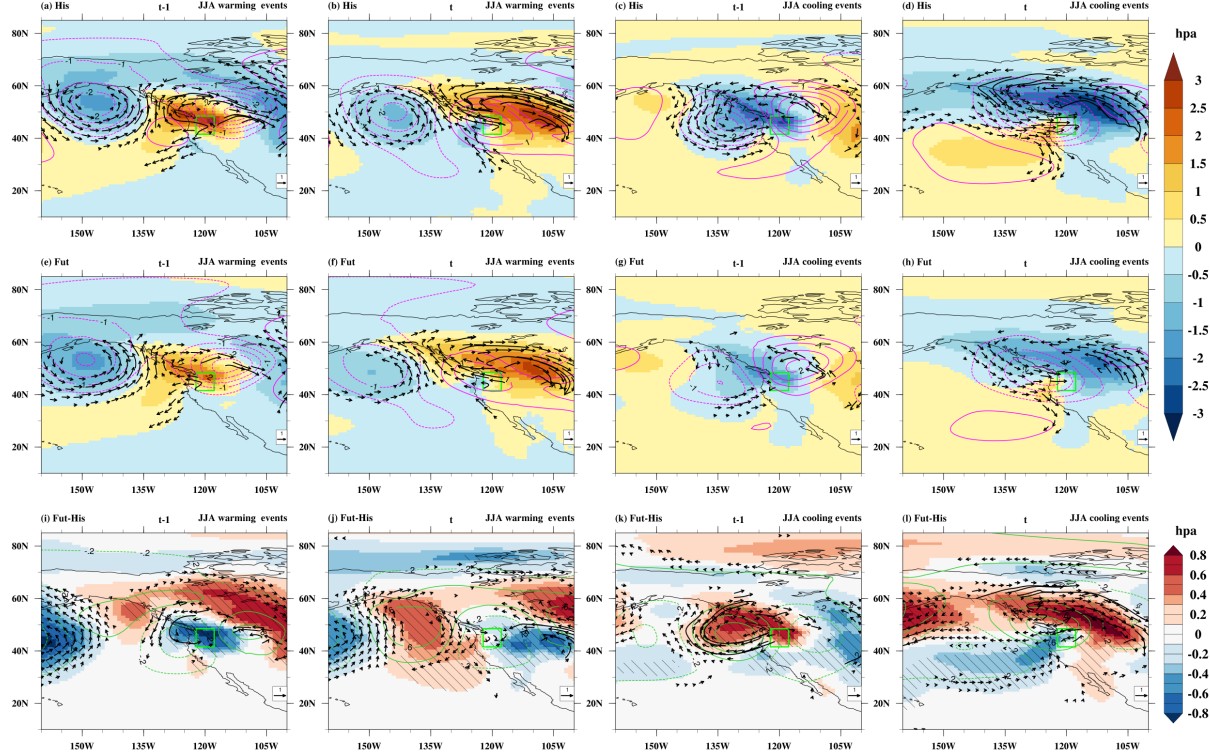


**Figure 7**. Composite of sea level pressure anomalies (hPa, color shading), wind anomalies at 850 hPa (**m s$^{-1}$**, vectors), and geopotential height anomalies at 500 hPa (gpm, magenta and darkgreen contours) relative to the seasonal mean on the **(a, e, i, c, g, k)** previous day (*t-1*) and **(b, f, j, d, h, l)** event day (*t*) of the warming **(a-b, e-f and i-j)** and cooling **(c-d, g-h and k-l)** events during June-August (JJA) in **(a-d)** historical climate (His), **(e-h)** future climate (Fut), and **(i-l)** projected future changes (Fut-His) at a selected grid box in western North America
(green box). Note that, in **(a-h)**, wind vector anomalies ≥ 1 **m s$^{-1}$** and in **(i-l)**, wind vector difference anomalies ≥ 0.5 **m s$^{-1}$** are plotted. The dotted and bold contours indicate negative and positive geopotential height anomalies, respectively. Additionally, the hatched area shows where the ensemble mean of sea level pressure differences exceeds the 95% confidence threshold based on a Student's t-test.

In the historical climate simulations, DTDT warming events are typically associated with a positive SLP anomaly near the
selected grid box at *t–1*, but with a cyclonic anomaly at 500 hPa with a maximum to the east (Figure 7a). Winds near the grid box are weak, and air masses originate in the vicinity, mostly over the continent (Figure 8a). By day *t*, the cyclonic anomaly at 500 hPa is replaced by a mature anticyclonic anomaly that began developing along the west coast on day *t–1* (Figure 7b). This transition is associated with the accumulation of warmer air masses over the continent beneath an amplified upper-level ridge (Fig. S7a and Figure 8b). Nevertheless, wind changes near the grid box between *t-1* and *t* are small, and air masses have
nearly identical source regions (Figures 8a-b) and source temperatures (Figure 8m). Accordingly, the contribution of advection to the warming is small and even slightly negative (–0.4°C); however, there is significant variability between individual events (Figure 9a). The warming is thus driven by adiabatic and diabatic processes during the three days of air mass transport, with increased adiabatic warming due to stronger subsidence within the anticyclonic circulation anomaly at *t* compared to *t–1,* contributing +2.4°C (Figures 8i, m, 9a). Near the surface, the descent weakens, allowing substantial diabatic warming likely





linked to surface sensible heat fluxes, which further elevate surface temperatures (Figure 8q) and contribute approximately +3.4°C to the temperature change. Overall, warm advection plays a negligible role for DTDT warming events during the historical period, which are instead driven by increasing descent and diabatic heating (Figure 9a). However, note that the roles of advection and adiabatic processes in CESM-LE are underestimated and reversed compared to those in ERA5 reanalysis data (Fig. S5c in part I). Nevertheless, according to ERA5, diabatic warming is the largest contribution to DTDT warming

events.

In the future climate scenario, the synoptic pattern resembles that of the historical climate, albeit with subtle changes. DTDT warming events are linked to smaller positive SLP anomalies and a weakened anticyclonic circulation over the West Coast at *t-1* (Figures 7e, i). This goes along with a further reduction in the transport of oceanic air masses and increases the presence of

northerly and localised air masses near the target grid point at *t-1* (Figure 8e). By day *t*, the anticyclonic anomaly centred over the grid point is also weaker (Figure 7f, j) and air mass origins shift slightly southward (Figure 8f). Although there is a slight shift towards warm air advection (Figures 8m, n), contributing approximately +0.7°C to the projected future change in DTDT magnitude, it does not compensate for the overall reduction (Figure 9c). The primary driver of the projected decline in DTDT warming is a significant decrease in adiabatic warming (–1.2°C) due to a smaller mean descent of air masses (by –11 hPa)

particularly on the day of the event (Figures 8i, j), which is linked to a weakening of the anticyclonic anomaly at 500 hPa described above (see again Figure 7j, Fig. S7c). Surface diabatic heating plays a secondary role. Since changes in heating are slightly greater on day *t–1* than on day *t*, it contributes an additional –0.3°C to the overall reduction (Figure 8r). Thus, the decline in adiabatic warming, combined with a modest reduction in surface diabatic heating, is the key factor decreasing the magnitude of DTDT warming events under future climate conditions (Figure 9c). However, when considering only the last

day before arrival, the changes in diabatic heating are the main factor weakening the DTDT warming (by –0.9°C), while adiabatic warming changes are weaker and of opposite sign (+0.3°C). This suggests that the process decomposition of DTDT warming events over western North America depends on the temporal limits of the analysis.





**Figure 8.** The spatial distribution of trajectory density initiated on the previous day (*t-1*) and the event day (*t*) is depicted for both June-August (JJA) warming and cooling events over western North America (green box). Color shading shows the air mass density (%) during the 3d before arriving at the target grid box for **(a-d)** historical climate and **(e-h)** projected future change. In **(e-h)**, stippling indicates areas where ≥80% of the ensemble members agree on the sign of change. The mean Lagrangian evolution of distinct physical parameters (pressure, temperature, and potential temperature) is shown along the air mass trajectories initialised on the previous and event days for historical/future extreme events (1st and 3rd columns) and projected future changes in extremes (2nd and 4th columns). Additionally, bold circles show where the ensemble mean differences at each timestep exceed the 95% confidence threshold based on a Student's t-test.



During historical DTDT cooling events, an anticyclonic circulation anomaly at 500 hPa over and to the east of the selected grid box at *t-1* day shifts further eastward and is replaced by an eastward-moving cyclonic anomaly from the Pacific Ocean on

the day of the event (Figure 7c-d). This is accompanied by an eastward shift of a negative SLP anomaly and stronger westerly winds on the southern flank of the cyclonic anomaly reaching the grid box on day *t*, such that the density distribution of air masses extends further into the western Pacific region compared to *t-1* (Figures 8c-d). As a result, a slight shift towards cold air advection leads to an average temperature drop of –1.6°C (Figures 8o, 9b). This advective cooling is partly offset by a modest increase in adiabatic warming (+1°C) due to stronger air mass descent on the day of the event (Figures 8k, o). However,

the main factor driving the DTDT decrease is reduced diabatic heating near the surface on day *t* relative to *t-1* (Figure 8s), mainly on the last day before the trajectories arrive (Figure 9b), with a mean contribution of –5.5°C. This pattern aligns with ERA5 reanalysis results, lending confidence to the CESM–LE representation of DTDT cooling events (see Fig. S5d in Part I).

In projected future DTDT cooling events, the configuration of circulation anomalies stays similar to present-day, but their

magnitude significantly weakens (Figures 7g-h, k-l and S7d). Air mass origins shift towards the continent at *t-1*, but northward and towards the ocean, with a reduction of continental airmass, on day *t* (Figures 8g, h). Together, this leads to a future reduction in the temperature difference between *t-1* and t at the air mass origin (Figure 8o, p), thereby reducing the contribution of cold air advection to the DTDT cooling by approximately +0.4°C (Figure 9d). The cooling is further suppressed by increased adiabatic warming (+0.6°C), linked to stronger mean ascent (by 6 hPa) at day *t* (Figures 8k, o). Near the surface, diabatic

cooling is projected to slightly increase in the future (by -0.2°C, Figure 8s). Therefore, future reductions in DTDT cooling events are mainly driven by reduced advection and further increases in adiabatic warming, with changes in diabatic cooling playing only a minor role (Figure 9d).



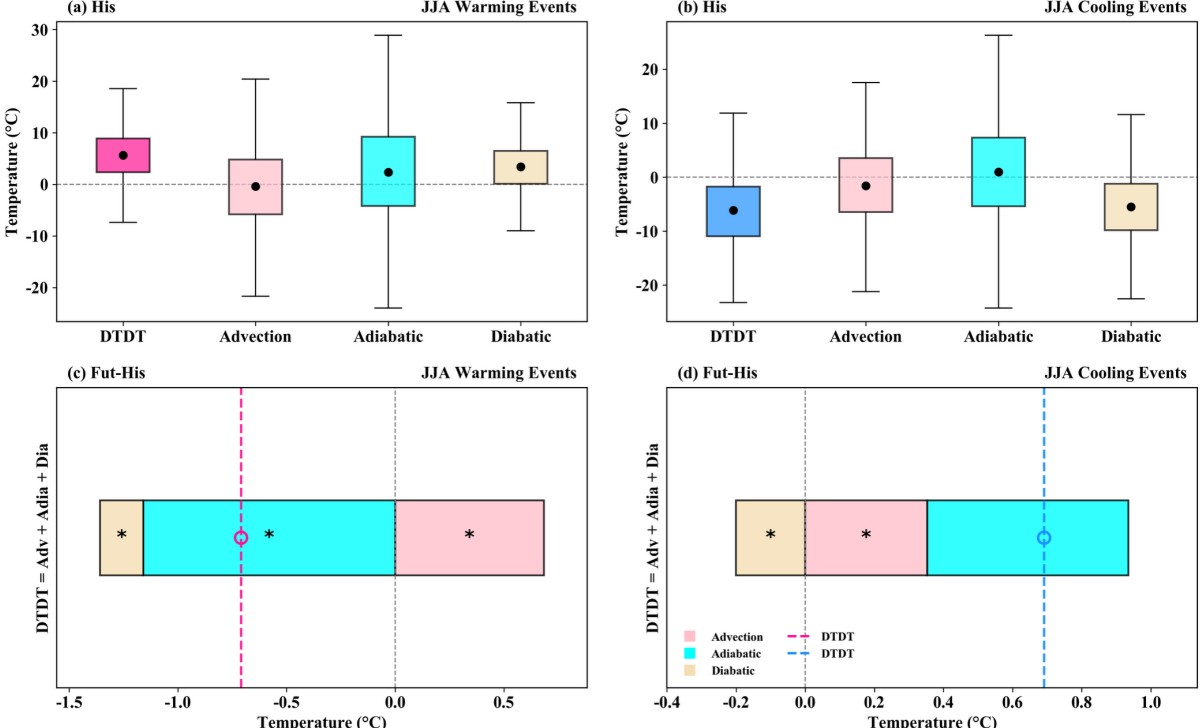

**Figure 9**. The contribution of the different physical processes (advection, adiabatic and diabatic temperature change) over western North America during June-August (JJA) to the genesis of DTDT **(a, c)** warming and **(b, d)** cooling events during historical climate **(a-b, box plots)** and projected future change **(c-d, stacked plots)** according to Eq. (2), which refers to a 3d-time scale. The box spans the 25th and 75th percentiles of the trajectory data; the black dot inside the box gives the mean of the related quantities in the historical climate, and the whiskers indicate 1.5 times the interquartile range in panels **(a)** and **(b).** The dotted lines in the stacked plots in panels **(c)** and **(d)** show the mean future change for DTDT warming and cooling events**,** respectively, and coloured bars indicate the contributions of the individual processes. Circle and * symbols mark future change distributions for which the ensemble mean differences exceed the 95% confidence threshold based on a Student's t-test.

### 3.2.2 Projected future intensification of extreme DTDT changes

During DJF, many tropical regions (Amazon, southern Africa, and Southeast Asia) are projected to experience an increase in extreme DTDT changes compared to the historical climate. To investigate the mechanisms behind extreme DTDT changes, we focus on a specific location in tropical South America (Figures 10-12; see figures S10-S11 for another location in Southern Africa).

In the CESM historical simulation, air masses at -3d cluster around the selected target grid box, indicating that local conditions mainly drive DTDT changes (Figure 10a-d), similar to the ERA5 analysis (see Figure 9a-d in Part I). During warming events, the temperature and pressure evolution along the backward trajectories is very similar for trajectories initialised at *t-1* and *t* between 3 and 1 days before reaching the target location (Figures 10i and m), while diabatic heating within the last 24 hours





plays a significant role for the DTDT increase (Figure 10q). Therefore, we analyse the physical processes on a one-day time scale (Figure 11a), in contrast to the 3d-analysis presented above for the extratropics. Composite precipitation and cloud cover

are further analysed in Figure 12 to explore these local diabatic effects. On day $t$-$1$, significant precipitation (5–8 mm d$^{-1}$) and high cloud cover (80-90 %) reduce solar radiation and thus diabatic heating, leading to lower temperatures. Conversely, on day $t$, decreased precipitation ($\leq$3 mm d$^{-1}$) and less cloud cover (70-80 %) enhance diabatic heating, contributing to higher temperatures (Figures 10m, 12b). This shift from wet and cloudy to dry and less cloudy conditions highlights the role of albedo changes and solar heating in driving a +2°C temperature increase over 1d, driven by enhanced diabatic heating (+2.2°C, Figure

11a). The magnitude of DTDT warming events in ERA5 and CESM-LE is nearly identical, resulting from a similar scale of diabatic heating (see also Figure 9k in part I and Figure 11a).

The localised patterns persist during future DTDT warming events, with high air particle densities at -3d continuing to cluster around the grid box and towards its east (reduced in the west) for day $t$-$1$. For day $t$, they are distributed even closer to the grid

cell (reduced in the south) compared to the historical climate (Figures 10e-f). Thus, changes in remote advection and adiabatic warming play only a minor role in future DTDT changes (Figures 10j, n and 11c). The projected warming is primarily driven by diabatic heating (nearly +1°C) near the surface over the last 24 hours, emphasising the enhanced role of radiative heating (Figures 10r and 11c). This finding is further supported by a significant increase in precipitation and cloud cover on day $t$-$1$, which, however, subsequently decreases on day $t$, contributing to the intensified DTDT warming in the future climate (Figures

12i-j). Also, for tropical Southern Africa (Fig. S10c) and Southeast Asia (not shown), the intensification of DTDT warming events is mainly driven by diabatic heating.





**Figure 10.** The spatial distribution of trajectory density initiated on the previous day (*t-1*) and the event day (*t*) is depicted for both December-February (DJF) warming and cooling events over tropical South America (green box). Color shading shows the air mass density (%) during the 3d before arriving at the target grid box for **(a-d)** historical climate and **(e-h)** projected future change. In **(e-h)**, stippling indicates areas where ≥80% of the ensemble members agree on the sign of change. The mean Lagrangian evolution of distinct physical parameters (pressure, temperature, potential temperature) is shown along the air mass trajectories initialised on the previous and event days for historical/future extreme events **(1st and 3rd columns)** and for projected future changes in extremes **(2nd and 4th columns).** Additionally, bold circles show where the ensemble mean differences at each timestep exceed the 95% confidence threshold based on a Student's t-test.



Also, for the historical DTDT cooling events, the distribution of air masses on both days is concentrated around the grid box, with remote advection and adiabatic heating playing little role for the temperature decrease (Figures 10c-d, k, o, s, 11b). These events are primarily driven by local diabatic effects near the surface in the 24 hours preceding the air masses' arrival at the

target location, with magnitudes comparable to those of the ERA5 reanalysis (see Figure 9l in Part I). Similar to DTDT warming events, on day *t-1*, precipitation (8–10 mm d$^{-1}$) and cloud cover (70-80 %) are smaller, leading to increased diabatic heating and higher temperatures, compared to the larger precipitation (>10 mm d$^{-1}$) and cloud cover (80-90%) on day *t*, which result in decreased diabatic heating and lower temperatures (Figures 11b and 12c-d). This transition from dry, less cloudy conditions to wet, cloudy conditions highlights the significant role of changes in albedo and solar heating in driving surface

diabatic cooling.

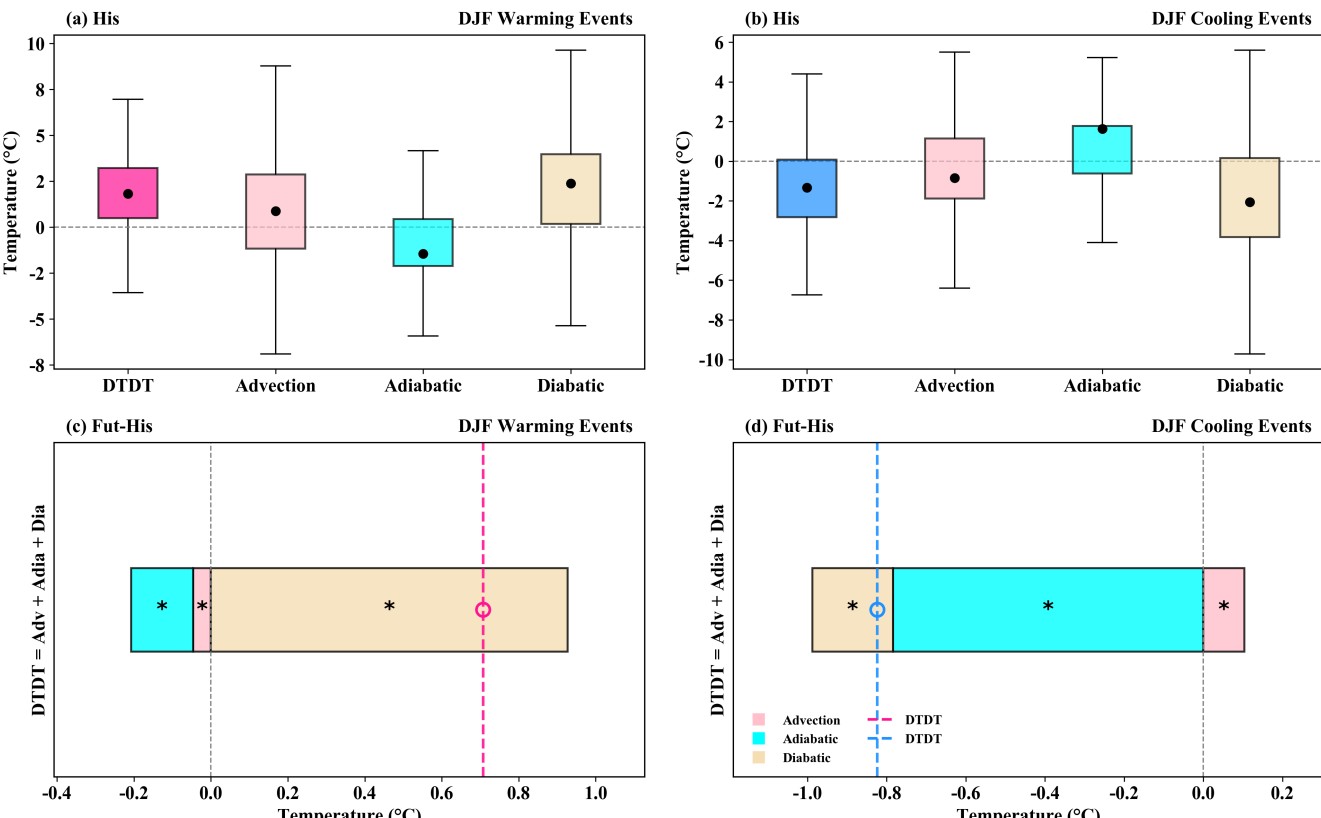

**Figure 11.** The contribution of the different physical processes (advection, adiabatic and diabatic temperature change) over tropical South America during December-February (DJF) to the genesis of DTD DTDT (**a, c**) warming and (**b, d**) cooling events during historical climate

(**a-b**, box plots) and projected future change (c-d, stacked plots) according to Eq. (2), which refers to a 1d-time scale. The box spans the 25th and 75th percentiles of the trajectory data; the black dot inside the box gives the mean of the related quantities in the historical climate, and the whiskers indicate 1.5 times the interquartile range in panels (**a**) and (**b**). The dotted lines in the stacked plots in panels (**c**) and (**d**) show the mean future change for DTDT warming and cooling events, respectively, and coloured bars indicate the contributions of the individual processes. Circle and * symbols mark future change distributions for which the ensemble mean differences exceed the 95% confidence

threshold based on a Student's t-test.



In future DTDT cooling events, local effects are projected to remain dominant, with increased air particle densities around and to the southwest of the grid box (reduced in the north) compared to the historical climate (Figures 10g-h). While the contributions of changes in advection to the future intensification of DTDT cooling are small, a reversal in the role of adiabatic warming is the primary factor for this intensification (Figure 11d). In contrast to the historical climate, in which a stronger

descent and adiabatic warming on day *t* compared to *t-1* slightly offsets the diabatically-driven cooling (Figures 10k, 11b), in the future, the ascent and adiabatic warming are projected to weaken on day *t*, leading to increased cooling (Figures 10l, p). In addition, an intensification of diabatic cooling (Figures 10t, 11d) associated with a marked decline in precipitation and cloud cover on day *t-1*, followed by an increase on day *t* (Figures 12k-1), further enhances DTDT cooling. A consistent pattern is also evident in Southeast Asia (not shown). A synergy between adiabatic and diabatic processes, albeit with a larger

contribution from diabatic changes, leads to future intensification of DTDT cooling events over South Africa (Fig. S10d).

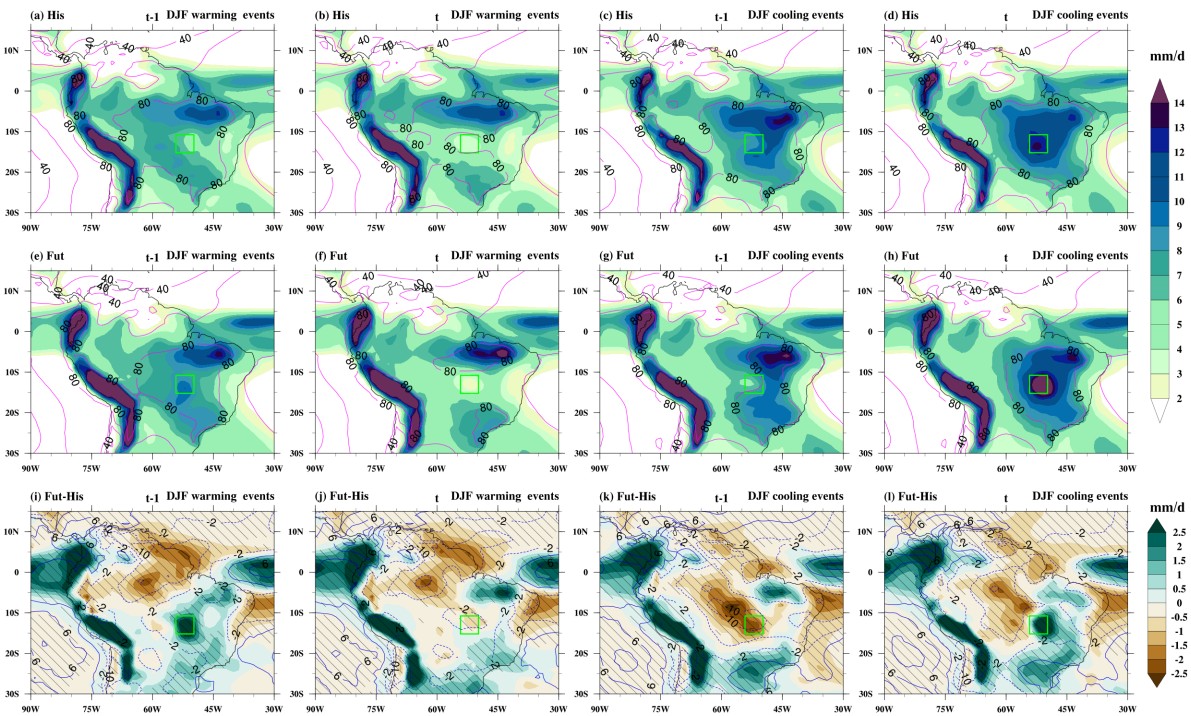

**Figure 12.** Composite of absolute total precipitation (**mm d$^{-1}$**, colour shading) and total cloud cover (%, magenta and blue contours) on the previous (*t-1*) and event (*t*) days of warming and cooling events during December-February (DJF) over South America. The top panel displays ensemble means for the (**a-d**) historical climate (His), (**e-h**) future climate (Fut), and (**i-l**) projected future change (Fut-His). The

green box represents the study grid box, and the hatched area in (**i-l**) indicates that the ensemble mean of the total precipitation differences exceeds the 95% confidence threshold based on a Student's t-test. The green and brown shading illustrate increases and decreases in total precipitation, respectively, while the magenta and blue bold and dotted contours represent increases and decreases in total cloud cover (in Figure **i-l**).



During JJA, some subtropical and midlatitude regions, such as the Sahel, parts of Europe, Southern Asia, Central America, and the Amazon, are projected to experience an increase in extreme DTDT changes relative to the historical climate. To study the underlying atmospheric circulation and physical processes, we select a grid box over central Europe (Figures 13-15). Further examples for locations in the subtropics (Figs. S13-14) and midlatitudes (Figs. S15-16) are shown in the supplement.

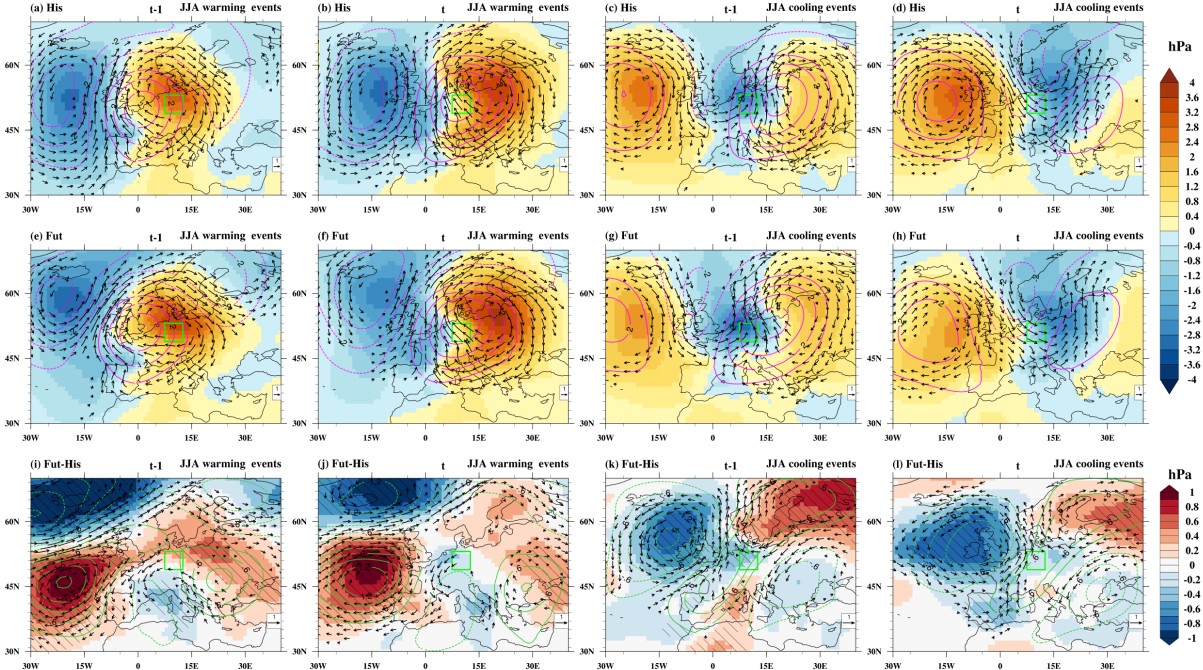

**Figure 13.** Composite of sea level pressure anomalies (hPa, color shading), wind anomalies at 850 hPa (**m s$^{-1}$** , vectors), and geopotential height anomalies at 500 hPa (gpm, magenta and darkgreen contours) relative to the seasonal mean on the **(a, e, i, c, g, k)** previous day (**t-1**) and **(b, f, j, d, h, l)** event day (**t**) of the warming **(a-b, e-f and i-j)** and cooling **(c-d, g-h and k-l)** events during June-August (JJA) in **(a-d)** historical climate (His), **(e-h)** future climate (Fut), and **(i-l)** projected future changes (Fut-His) at a selected grid box in central Europe (green box). Note that, in **(a-h),** wind vector anomalies ≥3 **m s$^{-1}$** and in **(i-l)**, wind vector difference anomalies ≥0.5 **m s$^{-1}$** are plotted. The dotted and bold contours indicate negative and positive geopotential height anomalies, respectively. Additionally, the hatched area shows where the ensemble mean of sea level pressure differences exceeds the 95% confidence threshold based on a Student's t-test.

During JJA warming in central Europe in the historical CESM-LE simulations, a mature trough–ridge anomaly pattern shifts eastward from day *t-1* to *t* (Fig. S12a; Figures 13a-b), bringing central Europe under the influence of southerly winds and warm continental air masses on the event day (Figures 14a–b), which aligns with the ERA5 synoptic pattern (see Figure 6a-c and Fig. S6a-b in Part I). This transition enhances warm air advection (Figure 14m), which plays a crucial role in driving the temperature increase, resulting in an average temperature rise of +3.2°C (Figure 15a). In contrast, adiabatic warming makes a minor negative contribution, averaging –0.3°C, due to slightly more substantial subsidence on day *t–1* than on day *t* (Figures 14i, m). However, the effect of adiabatic warming varies considerably across individual events, shown by the bars and whiskers



(Figure 15a). Enhanced diabatic heating contributes an average of +2.5°C to the DTDT increase, with a more pronounced rise in *θ* on day *t* than on day *t–1* (Figure 14q). Overall, warm air advection and diabatic heating are the primary contributors to DTDT warming events, with adiabatic warming playing a minor role in the historical climate (Figure 15a). A quantitative comparison reveals that CESM-LE indicates contributions from advection and diabatic heating similar to those of ERA5 (see
Figure 7k, part I).

In general, the synoptic-scale flow pattern associated with future DTDT warming events resembles the historical pattern (Figures 13e, f). However, notable differences are evident, particularly upstream over the North Atlantic (Figures 13i, j), where the low-pressure anomaly weakens. Simultaneously, the adjoining ridge pattern at 500 hPa over the continent shifts slightly
southeastward compared to the historical period. Together, these result in a decrease in the inflow of westerly, maritime air masses and a shift towards continental sources on both days involved in the DTDT warming (Figure 14e, f). These subtle changes in circulation patterns result in a modest increase in warm-air advection, contributing approximately +0.5°C to the DTDT change (Figure 15c). Meanwhile, vertical descent intensifies on day *t–1* (by ~5 hPa) but diminishes notably on day *t* (by ~10 hPa), leading to a negative adiabatic warming contribution that offsets the DTDT increase by –1.1°C (Figures 14j, n).
Conversely, diabatic heating increases on both days compared to the historical period, with a greater increase on day *t* (Figure 14r). This intensified diabatic heating adds +1.9°C to the DTDT change. Overall, the combined effect of slightly increased warm air advection and amplified diabatic heating—despite being partly offset by enhanced adiabatic warming—appears to be the main factor driving the projected future intensification of DTDT warming events (Figure 15c). A similarly important role of diabatic heating intensification is also evident in future DTDT warming events over the selected subtropical regions of
southern Asia and the Sahel (Figs. S13-14), as well as in Central America and northern Amazon (not shown). In contrast, changes in advection are more significant for events in southern South Africa, northern Asia (Figs. S15-16), and northern Europe (not shown).






**Figure 14.** The spatial distribution of trajectory density initiated on the previous day (*t-1*) and the event day (*t*) is depicted for both June-August (JJA) warming and cooling events over central Europe (green box). Color shading shows the air mass density (%) during the 3d before arriving at the target grid box for **(a-d)** historical climate and **(e-h)** projected future change. In **(e-h),** stippling indicates areas where ≥80% of the ensemble members agree on the sign of change. The mean Lagrangian evolution of distinct physical parameters (pressure, temperature, and potential temperature) is shown along the air mass trajectories initialised on the previous and event days for historical/future extreme events (1st and 3rd columns) and projected future changes in extremes (2nd and 4th columns). Additionally, bold circles show where the ensemble mean differences at each timestep exceed the 95% confidence threshold based on a Student's t-test.



During JJA cooling events in the historical climate, there is a transition from warm continental air masses to colder maritime air masses, coinciding with the development and eastward shift of a North Atlantic ridge (Figures 13c-d, 14c-d and Fig.S12b).

This cold air advection causes an average decline in surface temperature of −5.5 °C (Figures 14o, 15b). The cooling is partly offset by a modest increase in adiabatic warming (~0.7 °C), driven by stronger descent of air masses on day $t$ (Figures 14k, o). Additionally, reduced diabatic heating contributes a further temperature drop of −0.5 °C (Figure 14s). Overall, strong cold air advection and slightly reduced diabatic heating are the main drivers of DTDT cooling events over central Europe under historical conditions (Figure 15b), which also agrees with the ERA5 reanalysis (see Figure 7l in Part I). In the CESM-LE, the

contributions from advection and adiabatic warming are somewhat larger, but the diabatic contribution is similar to that in ERA5.

For projected future DTDT cooling events, on both days, the synoptic atmospheric circulation over the continent remains largely similar to that in the historical climate (Figures 13c-d, g-h, and Fig. S12d). However, changes are projected over the

North Atlantic, where the intensity of the high-pressure anomalies weakens at both the surface and 500 hPa (Figure 13k-l). Regarding the air mass origin, this is associated with an increase in westerly sources on both days, a reduction in southerly inflow at day $t$-$1$, and a reduction in northeasterly sources on day $t$ (Figure 14g-h). Together, this slightly reduces the contribution of horizontal temperature advection to the DTDT cooling (+0.3°C). Projected future changes in adiabatic warming are negligible (Fig. 14l, 15d), such that changes in diabatic heating are the primary contributor to the future DTDT cooling

intensification (–2.2 °C), which results from a much stronger intensification of the diabatic heating of air masses on day $t$-$1$ compared to day $t$ (Figures 14t, 15d). Changes in diabatic heating are also the main driver of the future intensification of DTDT cooling over Southern Asia, the Sahel, Northern Asia (Figs. S13-14, S16), as well as Central America, the northern Amazon, and Northern Europe (not shown). At the same time, all processes contribute more equally to events over southern South Africa (Fig. S15).




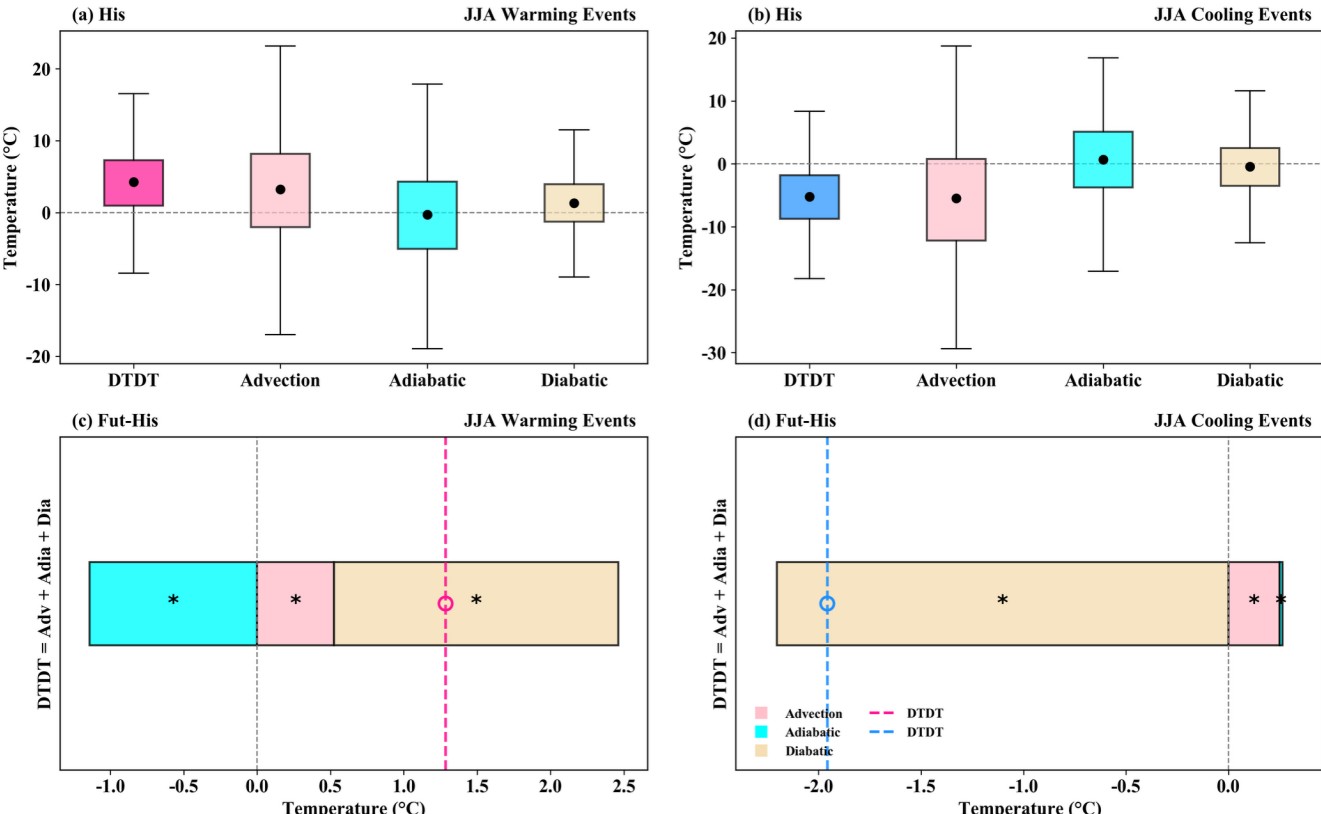

**Figure 15.** The contribution of the different physical processes (advection, adiabatic and diabatic temperature change) over central Europe during June-August (JJA) to the genesis of DTDT **(a, c)** warming and **(b, d)** cooling events during historical climate **(a-b, box plots)** and projected future change **(c-d, stacked plots)** according to Eq. (2), which refers to a 3d-time scale. The box spans the 25th and 75th percentiles of the trajectory data; the black dot inside the box gives the mean of the related quantities in the historical climate, and the whiskers indicate 1.5 times the interquartile range in panels **(a)** and **(b)**. The dotted lines in the stacked plots in panels **(c)** and **(d)** show the mean future change for DTDT warming and cooling events**,** respectively, and coloured bars indicate the contributions of the individual processes. Circle and * symbols mark future change distributions for which the ensemble mean differences exceed the 95% confidence threshold based on a Student's t-test.

### 4. Discussion and Summary

This study has examined historical patterns and projected future changes in extreme DTDT variations and their underlying physical processes based on the CESM-LE, a large single-model ensemble designed to understand Earth system variability and global change (Kay et al., 2015). Our results from the CESM-LE in the historical climate indicate that DTDT variations and extremes are more pronounced in the extratropics than in the tropics during both DJF and JJA, consistent with the ERA5 reanalysis (see also Figures 1-3 in Part I). A comparison of CESM-LE and ERA5 reanalysis results reveals notable regional differences, particularly an overestimation in mid- to high latitudes and Southeast Asia during DJF, and in some subtropical areas during JJA. The larger magnitude of $\sigma_{DTDT}$ in CESM-LE is due to both higher daily standard deviations and lower autocorrelation compared to ERA5 (Fig. S1). The differences in daily standard deviations align with patterns observed in other models from the CMIP5 and CMIP6 initiatives (Bathiany et al., 2018; Giorgi & Raffaele, 2022). Simpson et al. (2022) argue





that, in some regions, they are partly due to the limited representation of snow density, which affects the thermal conductance
of the snow layer. Nevertheless, the general agreement between the spatial patterns of temperature variability in the CESM-
LE and ERA5 increases confidence in future projections.

The projected future DTDT variations and extremes reveal distinct seasonal and spatial differences. During DJF, projected
DTDT changes follow a dipole pattern, characterised by weakening in mid- to high latitudes and intensification in the tropics.
Also, during JJA, most tropical regions exhibit a significant intensification, but the signal in the extratropics is less coherent,
with only a few regions, such as Greenland, western North America, eastern North America, and southern South America,
experiencing a weakening in the magnitude of DTDT change. These results are similar to those of other recent studies on
DTDT variations and extremes (Liu et al., 2025; Wang et al., 2025; Xu et al., 2020; Zhou et al., 2020). However, we further
show how the changes in DTDT variations ($\sigma_{DTDT}$) can be decomposed into contributions from changes in the standard
deviation of daily mean temperature ($\sigma_T$) and its autocorrelation ($r_{1,T}$). In particular, the projected $\sigma_{DTDT}$ changes are mainly
driven by $\sigma_T$ changes during DJF and, in most regions, also during JJA. Furthermore, the projected $\sigma_T$ changes in CESM-LE
are generally similar to those in other CMIP5 models (Bathiany et al., 2018; Tamarin-Brodsky et al., 2020). However, in some
regions, changes in temporal autocorrelation also affect the projected $\sigma_{DTDT}$ change, even when the $\sigma_T$ change is small. For
example, the projected increase in $\sigma_{DTDT}$ over western Africa in DJF is due to a reduction in $r_{1,T}$, whereas a decrease in $\sigma_{DTDT}$
over Greenland and northwestern North America in JJA is linked to an increase of $r_{1,T}$. A similar future increase in
autocorrelation over longer time scales is projected across western North America by other models (Li & Thompson, 2021),
which may be linked to slow-moving weather patterns and the Arctic amplification (Kornhuber & Tamarin-Brodsky, 2021).
These results indicate the need to study projected future changes in daily persistence further using a multi-model ensemble.

Additionally, we have examined in detail the physical mechanisms driving regional extreme DTDT changes during DJF and
JJA using a combination of Eulerian composites and a Lagrangian temperature decomposition into contributions from
advection, adiabatic and diabatic temperature changes. This detailed process analysis goes substantially beyond prior studies
on extreme DTDT variations (Liu et al., 2025; Zhou et al., 2020). The physical processes observed in CESM-LE for historical
simulations are generally similar to those in ERA5 (see also Part I), thereby increasing confidence in future projections. The
future weakening of both types of extremes (DTDT warming and cooling) over the extratropics during DJF is mainly due to a
reduced contribution of advection, consistent with previous studies on general aspects of temperature variability (Tamarin-
Brodsky et al., 2020; Wang et al., 2019; Wang et al., 2025). These changes are driven by a combination of subtle shifts in
circulation patterns and changes in temperature gradients associated with Arctic amplification and changes in land-sea contrast,
leading to a decrease in DTDT variance (Chen et al., 2019; Dai & Deng, 2021; Garfinkel & Harnik, 2017; Screen, 2014; Zhou
et al., 2020). For example, during future DTDT warming events over North America, the synoptic flow anomalies propagate
faster compared to the historical period, with stronger southwesterly flow on day *t-1* than on day *t*. In addition, the projected
temperature increase is larger at the still more poleward air mass origins on day *t-1* than on day *t*, most likely associated with



Arctic amplification. Both these factors reduce the DTDT variance. While advective changes predominate for these DTDT extremes, a minor contribution also arises from modifications in diabatic and adiabatic processes. Changes in diabatic processes, especially significant at the surface, are likely driven by variations in surface albedo and heat fluxes (Chen et al., 2019; Diro & Sushama, 2020).

The weakening of extreme DTDT changes during JJA shows clear regional and event-type differences, unlike the more uniform pattern in DJF. This trend is driven not only by changes in advection but also by significant contributions from both adiabatic and diabatic processes. During warming events, eastern North America experiences combined reductions in advection and diabatic heating, while western North America is mainly affected by a decrease in adiabatic warming. Similarly, cooling events show decreases in advection, along with region-specific changes: increased diabatic heating in the east and adiabatic warming in the west. Generally, in the Northern Hemisphere, the advection-driven changes associated with JJA extremes are likely linked to Arctic warming during summer as well (Coumou et al., 2018; Kornhuber & Tamarin-Brodsky, 2021). The changes in diabatic processes are probably associated with changes in surface net radiative forcing, as discussed by (Wang et al., 2025). Notably, despite the decline in extreme DTDT changes, daily heat extremes and heatwaves are still expected to intensify in the future, likely driven by intensified large-scale diabatic heating (Bartusek et al., 2022; Heeter et al., 2023; White et al., 2023; Zhang et al., 2023). Overall, the reduction in extreme DTDT variability across extratropical regions in JJA results from a complex interaction of dynamical and thermodynamic factors, whose relative importance varies across hemispheres, regions, and event types.

Conversely, an intensification of extreme DTDT variations is also evident in certain extratropical regions, particularly Northern Europe, Northern Asia and the south of Southern Africa (Figs. S15-16), where recent research indicates generally rising risks associated with temperature extremes (Beobide-Arsuaga et al., 2025; Ciavarella et al., 2021; Nangombe et al., 2019). In the southern hemisphere, increased DTDT extremes result from the combined influences of advection, adiabatic, and diabatic processes, whereas changes in the northern high latitudes are primarily diabatic, with advection changes playing some role (mainly during warming events), while vertical motion remains largely unaffected. Also, the projected intensification at lower latitudes primarily results from enhanced diabatic processes, with advection playing a secondary role (Figs. S13-14). One important mechanism by which diabatic processes can influence temperature variability is through land-atmosphere interactions (Beobide-Arsuaga et al., 2025; Sato & Nakamura, 2019), leading to amplified temperature fluctuations and more frequent extremes, including extreme DTDT changes (Cattiaux et al., 2015; Liu et al., 2025). For example, the future amplification of DTDT warming events in central Europe is driven by increased diabatic heating on the event day, which closely resembles the mechanisms underlying the intensification of heatwaves (Schielicke & Pfahl, 2022). Heatwaves often occur in parallel with soil moisture depletion, which lowers the latent heat flux and increases the sensible heat flux through a higher Bowen ratio (Lin et al., 2022; Zscheischler & Seneviratne, 2017). Overall, these insights imply that regions such as the



Sahel and central Europe could experience more pronounced soil drying in a warming climate (Elkouk et al., 2021; Ruosteenoja et al., 2018), possibly creating a feedback loop that amplifies future temperature extremes.

Similarly, an intensification in extreme DTDT changes is projected in the tropics during DJF, highlighting increased risks for many developing countries. These regions are expected to encounter a disproportionate share of climate change's socioeconomic, agricultural, and health impacts (Bathiany et al., 2018; Ebi et al., 2025; Linsenmeier, 2023; Raymond et al., 2020). Historical and future extreme DTDT changes over these areas are not driven by large-scale advection, but by local diabatic and adiabatic processes. The intensification of future DTDT warming events is mainly caused by stronger local diabatic heating, linked to reduced cloud cover and precipitation on event days, which allows more solar radiation to reach the
surface and raise temperatures—similar to processes during heatwaves (Birch et al., 2022; McKinnon et al., 2024; Moustakis et al., 2020). For future DTDT cooling, local diabatic cooling remains relevant, whereas reduced adiabatic warming intensifies it. Nevertheless, such adiabatic changes and their influence on near-surface temperature might be linked to cloud-diabatic effects through lower-tropospheric static stability (Luo et al., 2024).

Our study is the first to apply a Lagrangian backwards-trajectory method to investigate the physical processes underlying projected future extreme DTDT changes. The results reveal clear seasonal and regional variations in the occurrence of these extremes, for which not only advection but also changes in adiabatic and diabatic processes are important. In the extratropics during DJF, reductions in extreme DTDT changes are mainly attributable to weaker temperature advection, primarily linked to Arctic amplification. Conversely, the weakening of extremes during JJA is primarily driven by diabatic and adiabatic
processes, with some changes in advection, especially in North America. In contrast, there is substantial intensification of DTDT extremes over tropical and subtropical land areas during JJA, mainly driven by diabatic processes that are likely associated with changes in surface fluxes. Adiabatic warming changes, combined with diabatic processes, are crucial for the increase of extremes in the tropics during DJF. These findings highlight the importance of accounting for the Lagrangian temperature change of air masses when examining spatial and seasonal variations in future DTDT changes, which can
significantly affect ecosystems, public health, and infrastructure. They also emphasise the need for region-specific adaptation strategies to mitigate the risks associated with rapid temperature changes.

**Code and data availability**

The code for the trajectory model LAGRANTO is available at https://iacweb.ethz.ch/staff/sprenger/lagranto/ (Sprenger & Wernli, 2015). The model code for CESM version 1 used for the ensemble simulation is available from (Kay et al., 2015);
https://www2.cesm.ucar.edu/models/cesm1.0/. ERA5 data are available via the Copernicus Climate Change Service (Hersbach et al., 2020); https://doi.org/10.24381/cds.143582cf.



**Author contributions**

Both authors designed the study. KH performed the analysis, produced the figures, and drafted the manuscript. Both authors discussed the results and edited the manuscript.

**Competing interests**

Stephan Pfahl is executive editor of WCD.

**Acknowledgments**

We acknowledge the HPC service of ZEDAT, Freie Universität Berlin, for providing computational Resources (Bennett et al., 2020).

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
