# Peer review of "Physical Processes Leading to Extreme Day-to-day Temperature Change - Part II: Future Climate Change"

_EGUsphere, 2025_

## Referee Comment (RC1)

**Review of "Physical Processes Leading to Extreme Day-to-day Temperature Change - Part II: Future Climate Change"**
by Kalpana Hamal and Stephan Pfahl
submitted to Weather and Climate Dynamics

**General comment:**

In this paper, the authors investigate extreme day-to-day temperature changes and how these are projected to evolve under future climate conditions. They analyse an ensemble an ensemble of simulations with a climate model, comparing results for the present climate with ERA5 reanalysis data to build confidence in the model performance, before examining projected future changes. The strongest projected changes in day-to-day temperature variability are found for DJF, with decreases in the extratropics and increases in the tropics. The authors further analyse associated changes in flow patterns and in the contributions from adiabatic warming, diabatic heating, and advection by accumulating the respective terms along computed backward trajectories. They identify a reduced contribution from advection as the main driver of decreasing day-to-day temperature changes in the extratropics in DJF, while increases in the tropics are largely attributed to changes in adiabatic and diabatic warming.

Overall, I find the study interesting and the results valuable. In particular, the successful implementation of trajectory calculations within a climate model framework is a notable strength of the paper, as such analyses are often challenging due to the limited temporal and spatial resolution of typical climate model output.

That said, I believe the presentation of the material could be substantially improved. My comments below are therefore mainly concerned with the clarity and structure of the presentation, rather than with the underlying scientific approach or results.

**Major comments:**
1) The descriptions in the manuscript are often quite lengthy and at times somewhat repetitive. For example, results are first presented for DJF warming events in the present and future, then DJF cooling events, followed by JJA warming and cooling events, and subsequently for the tropics. To enhance reader engagement, it might be helpful to introduce the figures once and then focus more on highlighting the key differences, while keeping the descriptions concise and emphasizing the essential points.

2) Playing the devil's advocate: How physically meaningful are the computed trajectories within the atmospheric boundary layer and especially in the tropics where turbulent mixing is intense? I think a brief discussion on the limitations or uncertainties associated with the trajectories in these regions would strengthen the study.

**Minor comments:**
L15/16: The mention of a "clear dipole pattern" is somewhat confusing to me. You mention a "clear dipole pattern", but then for JJA the pattern does not clearly take a dipole form. Consider rephrasing.

L18: "only" instead of "also"?

L19: I think you should be more careful here when writing "due to Arctic Amplification".

L37: "imperative" is a very strong word. I would be a bit more moderate here.

L56: "for the past" instead of "in the past"?

L63: I would avoid citing Mayer (2025) when discussing the importance of diabatic heating, as Mayer (2025) emphasizes the role of advection in temperature extremes rather than adiabatic or diabatic processes.

L69: "process understanding" instead of "processes understanding"?

L95: "use" instead of "utillise"

L106: In its current position, the formula does not appear to be well integrated into the text. The same applies to Eq. (2).

L110: Why are these seasons "key seasons"? Rather explain or omit the "key".

L124-127: I would omit the lengthy description of all the individual grid points in the supplement.

Eq. (2): It might be helpful to write out the integrals explicitly to clarify exactly which terms are being computed.

Eq. (2): Why do you accumulate over 3 days? Is there a physical reason? Have you tested the sensitivity of your results to other accumulation periods?

L134-136: It is not clear what is meant by "mean temperature difference" or "mean adiabatic compression." Consider clarifying the meaning of "mean" here.

L147/148: There seems to be a contradiction: first, the results are said to fit ERA5 "in many regions," then to deviate "in large parts." Consider clarifying this.

Figure 1, 2, 3, etc.: I think it would be helpful to use white color for small deveations around 0.

L162: The pattern does not appear as "distinct" to me.

L166/167/177: Phrases like "changes are driven by", "increases due to", or "influence" imply causality to me. Since the decomposition (into $\sigma_T$ and autocorrelation) is "just" descriptive, I think you avoid implying causality.

L179/180: The statements about Chile are contradictory: first mentioning it as an exception, then saying "(apart from Chile)." Consider clarifying.

Figure S3: The description is very detailed, but the figure is in the supplement. Consider either shortening the description or moving the figure to the main text.

Figure 3/4/7/12/13: The figures are very small, which makes it difficult for the reader to fully appreciate their content.

L208: omit the "future" as projected already implies future?

L239: Out of curiosity: Do you have an idea why the contribution of the diabatic heating is larger in the CESM-LE compared to ERA5? Could this relate to vertical resolution?

L245: When mentioning "Rossby wave propagation," consider providing supporting evidence or omitting the comment.

L253: I was stumbling across the formulation "This reduction is because ..."

L585: "driven by" and "due to" as before. I think you should refrain from implying causality here.

Several transition words (e.g., "conversely" in L627, "however" in L583, and "in contrast" in L660) do not seem appropriate in their current context and may be misleading. Revisiting these connectors could help improve clarity and flow.

---

## Referee Comment (RC2)

**Review of "Physical Processes Leading to Extreme day-to-day Temperatures Changes, Part II: Future Climate Change"**

The work of Hamal and Pfahl is a continuation of their similar work using ERA5. They present a decomposition of the processes leading to extreme day-to-day temperature changes in a climate model (CESM), comparing a present period and a future period under global warming for extreme temperature changes in winter and summer. They provide a detailed analysis of these extreme temperature changes for several extratropical and tropical regions, looking at the synoptic conditions and especially decomposing the physical processes leading to the change of the temperature anomaly using a lagrangian backward trajectories analysis.

The paper is clear and well-written (although some parts have extensive descriptions of the atmospheric circulation that could be shortened in my opinion). I was already a reviewer in the first part of this article and I have nothing new to add to the methods, which are essentially similar. I have some technical comments below, including some statistical significance computations that should be done differently in my opinion, but apart from those I would be happy to recommend the paper after some revisions.

The only main limitation that I see is that I find the paper very descriptive without testing any physical theory. In other words, it could have been interesting to explore how the changes you see fit with some physical expectations for how those mechanisms are supposed to evolve. It is mentioned several times that Arctic amplification, and the associated change in the temperature gradient and general circulation, is expected to decrease the importance of advection and I think this kind of reasoning could be interesting to investigate further. I do not think this is a reason to reject the paper, but it could really add something on top of those descriptive mechanisms.

**Major comments**

1. Figure 1 and corresponding: the way the stipplings are computed does not look like a proper statistical test to me.
   a. If I understood correctly, for the present the authors flag as "significant" the grid points where 80% of members are within +/- 10% of the EAR5-derived respective quantities. I do not think this is correct: first the +/-10% for ERA5 is an ad-hoc measure of the uncertainty. Second, I do not see why 80% of the members should be a correct measure of a significant difference. I suggest to do an actual statistical test with a standard reference level of 95% significance for example. In essence you want to know whether the climate of CESM is compatible with the value for ERA5: that is what you need to test for. The climate of CESM is defined by all the members being put together: what you need to test is whether the sigma_DTDT, sigma_T and r_1,T of this climate are compatible with the same values for ERA5 (which also has an uncertainty). You could for example employ a bootstrapping approach on the ERA5 data (the values for the model are likely very well estimated given the amount of members you have) and check whether the distribution of values you obtain are compatible with the one from the model at the 95% level.
   b. Same for the future: why don't you simply test whether there is a significant difference in sigma_DTDT, sigma_T and r_1,T between the two climates by putting all members together in each period ?
2. For all your significance maps: you need to take into account correlations in statistical testing and employ a false discovery rate, see Wilks (2016).
3. Several times the authors argue that the model is doing a reasonable job in reproducing the statistics of ERA5. I am not sure this is so much the case, as exemplified by Figure 1 for example where the stipplings do not really cover most of

the regions (modulo my main comment 1). I think you should emphasize the differences more, including quantifying them when possible. One point for example is that the model seems to have a diabatic contribution larger than ERA5, which is something also found recently by Röthlisberger et al. (2025) in a different context: it seems to me that the model may be right for the wrong reasons.

**Minor comments**

1. Please precise which version of CESM you are using.
2. Figure 1: because of the strong meridional differences, you could plot the changes in the second column in percentage rather than absolute values.
3. Figure 3: the sigma_DTDT should be delta_T ?
4. Figure 4: the stipplings are barely visible.
5. Figure 5: I would suggest to scale the temperature and pressure differences by a global/regional warming level to see what is changing beyond the expected local warming.
6. Figure 6 and similar: maybe you could add the boxplots for the future on panels a and b also to compare the spread in each period (I do not expect the spread to be small, thus that the changes you observe are probably much smaller in intensity compared to the spread between events in each period).
7. Figure 7: why did you decide to change the position of the box for looking at extreme DTDT changes compared to Figure 4 ?

**References**

Wilks, D. (2016). "The stippling shows statistically significant grid points": How research results are routinely overstated and overinterpreted, and what to do about it. *Bulletin of the American Meteorological Society*, *97*(12), 2263-2273.

Röthlisberger, M., Sprenger, M., Beyerle, U., Fischer, E. M., & Wernli, H. (2025). Advective, adiabatic and diabatic contributions to heat extremes simulated with the Community Earth System Model version 2. *EGUsphere*, *2025*, 1-32.